# Preliminary Screening of South African Plants for Binding Affinity to the Serotonin Reuptake Transporter and Adenosine A₁/A₂ₐ Receptors

Andisiwe Mnqika [1], Adeyemi O. Aremu [2,*], H. D. Janse van Rensburg [3] and Makhotso Lekhooa [1,*]

1   DSI/NWU Preclinical Drug Development Platform, Faculty of Health Sciences, North-West University, Private Bag X6001, Potchefstroom 2521, South Africa; 43611753@mynwu.ac.za
2   Indigenous Knowledge Systems Centre, Faculty of Natural and Agricultural Sciences, North-West University, Private Bag X2046, Mmabatho 2790, South Africa
3   Centre of Excellence for Pharmaceutical Sciences, Faculty of Health Sciences, North-West University, Private Bag X6001, Potchefstroom 2521, South Africa; 23551917@mynwu.ac.za
*   Correspondence: oladapo.aremu@nwu.ac.za (A.O.A.); makhotso.lekhooa@nwu.ac.za (M.L.); Tel.: +27-18-389-2573 (A.O.A.); +27-18-299-2270 (M.L.)

**Abstract:** In South African traditional medicine, *Gomphocarpus fruticosus* (L.) W.T. Aiton, *Hypoxis hemerocallidea* Fisch. & C.A. Mey., and *Leonotis leonurus.* (L.) R.Br. have been recorded among different ethnic groups to be a valuable herbal remedy for the management of depression-related conditions. The current study investigated the affinity of these three plants toward the serotonin reuptake transporter (SERT) and adenosine A₁/A₂ receptors. Six solvents (water, methanol, acetone, dichloromethane, petroleum ether, and hexane) were used to extract the selected plants. We established that eight extracts exerted potential affinity based on the applied in vitro binding experiment. The methanol and acetone extracts of *Hypoxis hemerocallidea* had 60% specific binding of [³H]citalopram, an indication that almost 40% of the plant extracts were bound to the SERT. For the adenosine receptor binding assays, methanol and hexane extracts of *Leonotis leonurus* were the most active, with $rA_1K_i$ values of 0.038 and 0.176 mg/mL, respectively. In addition, the dichloromethane extract of *Gomphocarpus fruticosus* had an $rA_1K_i$ value of 6.46 mg/mL. Extracts from the more polar solvents methanol and dichloromethane had higher binding affinity. Additionally, these plant extracts acted as antagonists at the adenosine A₁ receptor. Overall, the current findings provide an indication of the potential antidepressant effects of some of the tested extracts based on their binding to the receptors evaluated. However, a combination of other in vitro assays is needed to establish possible mechanisms of action. In addition, computational analysis and profiling of plant extracts is crucial to identify the bioactive compounds with a higher affinity to the receptors. Ultimately, in vivo studies remain essential to allow for an in-depth elucidation of the mechanisms of action.

**Keywords:** biodiversity; depression; GTP shift assay; radioligand assay; traditional medicine

## 1. Introduction

Worldwide, depression affects more than 300 million people and an estimated 6–10% of the population will experience a depressive episode in any given year [1]. In South Africa, an expected 9.8% of the adult populace experience major (clinical) despondency sooner or later in their life [2]. Some of the frequently prescribed medications for depression include selective serotonin reuptake inhibitors (SSRI) which are known to exert antidepressant effects by blocking the serotonin transporter (SERT), and consequently the uptake of serotonin from the synaptic cleft, and in so doing, increase serotonin concentration in the synapse [3,4]. Adenosine A₁ and A₂ₐ receptors play a vital role in the brain by regulating the release of neurotransmitters [5,6]. Generally, A₁ receptor stimulation induces synaptic depression by reducing neurotransmitter release [7], whereas A₂ receptors are associated with increasing neurotransmitter release [8]. Existing animal studies

have suggested the inhibition of adenosine $A_{2A}$ receptors produces antidepressant-like behaviors, and adenosine receptor antagonists have been found to reverse the adenosine-mediated "depressant" effect [9,10]. Even though low- and middle-income countries account for 80% of the depression burden, only 10% of these patients have access to effective treatments [11].

About 11% of the drugs on the World Health Organization (WHO) list of essential medicines were derived from plant and/or natural products [12]. This further confirms the potential of plants as a source for drug development for either herbal medicines and/or single-molecule drugs. The WHO recommends evidence-based research to support the utilization of medicinal plants [12]. Globally, several plants have been identified and studied for their antidepressant activities [13,14]. For instance, *Withania somnifera* (L.) Dunal and *Hypericum perforatum* L. were tested in preclinical and clinical studies [15]. South Africa is rich in plant species, some of which are traditionally used to manage mental health. One of the most studied South African plants with antidepressant effects is *Mesembryanthemum tortuosum* L. (synonym: *Sceletium tortuosum*), which in a dried state is smoked, chewed, and inhaled as a snuff to provide a calming effect [16]. However, there are several other South African plants known to be utilized for mental-related conditions among different ethnic groups, but scientific evidence on their efficacy is limited [17]. Screening South African medicinal plants for affinity toward the SERT and adenosine receptors is the initial step in an a'ttempt to evaluate their antidepressant effects. Therefore, the current study investigated the potential antidepressant effect of three South African plants that were extracted using different solvents. In addition, the type of activity (agonist, inverse agonist, or antagonist) at adenosine $A_1$ receptors for the selected plant extracts was evaluated. The three plants investigated were *Gomphocarpus fruticosus* (L.) W.T. Aiton, *Hypoxis hemerocallidea* Fisch. & C.A. Mey., and *Leonotis leonurus* (L.) R.Br., which are well documented in ethnobotanical data to be utilized as remedies for depression-like conditions among different ethnic groups in South Africa [17].

## 2. Materials and Methods

### 2.1. Selection of Plants for Screening

Based on a previous study [17], 186 plants were identified as the botanical remedies used among different ethnic groups for depression-like ailments. The study revealed that a significant portion (82%) of the identified plants lack pharmacological evidence related to their antidepressant effects. On this basis, three plants with evidence of ethnobotanical use among at least two South African ethnic groups were selected for investigation.

### 2.2. Plant Material Collection

The leaves of *Leonotis leonurus* and corm of *Hypoxis hemerocallidea* were collected from the University of KwaZulu-Natal Botanical Garden. Following positive confirmation, voucher specimen numbers for *Leonotis leonurus* (NU0094474) and *Hypoxis hemerocallidea* (NU0094473) were deposited at the University of KwaZulu-Natal, South Africa (Table 1). The leaves of *Gomphocarpus fruticosus* were collected from the North-West University Botanical Garden. *Gomphocarpus fruticosus* was identified by P. Naidu, plant taxonomy research assistant, North-West University and allocated with voucher specimen number PUC0016056. The three plants were washed with distilled water and oven-dried at 37 °C for 2 days.

**Table 1.** Plant family, species, local names, parts traditionally used, and voucher numbers of materials collected for screening. (A)-Afrikaans; (E)-English; (S)-Sesotho; (X)-Xhosa; (Z)-Zulu.

| Family | Species | Local Name | Voucher Number | Plant Part use in Traditional Medicine |
|---|---|---|---|---|
| Apocynaceae | *Gomphocarpus fruticosus* (L.) W.T. Aiton | Milkweed (E); Tontelbos (A); Lebejana (S); Umsinga-lwesalukazi (Z) | PUC0016056 | Leaves |
| Hypoxidaceae | *Hypoxis hemerocallidea* Fisch. & C.A. Mey | Star flower, yellow star (E); Streblom (A); Inkomfe (Z); Lotsane (S) | NU0094473 | Corm |
| Lamiaceae | *Leonotis leonurus* (L.) R.Br. | Lion's ear, wild dagga (E); Wildedagga (A); Imvovo (X); Umcwili (Z) | NU0094474 | Leaves |

### 2.3. Plant Extraction

The dried leaves of *Gomphocarpus fruticosus* and *Leonotis leonurus* and sliced corm of *Hypoxis hemerocallidea* were ground to powder using a blender and stored at 4 °C until analysis. Using a 1:10 ratio, the ground plant materials were extracted with water, methanol, acetone, dichloromethane, hexane, or petroleum ether for 60 min at 23 °C facilitated by an ultrasonicator (ScienTech). The extraction was repeated and the extracts were filtered using a Buchner filtration system. This was evaporated using a rotary evaporator and freeze-dried until ready for use. The extracts were resuspended at a concentration of 100 mg/mL in dimethyl sulfoxide (DMSO) for the in vitro analysis.

### 2.4. Membrane Preparation

The collection of tissue samples for the assays was approved by the Faculty of Health Science Animal Research Ethics Committee of the North-West University (application number NWU-00780-22-A5). Sprague Dawley rats whole-brain membranes (including striata and excluding cerebellum and brain stem) were used for $A_1$ and SERT, and rat striatal membranes were used for the $A_{2A}$ radioligand binding assays, respectively. All procedures were carried out at 0–4 °C. The tissue was disrupted for 90 s (whole brain) or 30 s (striata) with the aid of a Polytron homogenizer (PT 10-35 GT) in ice-cold 50 mM Tris buffer (pH 7.7 at 25 °C). The resulting homogenate was centrifuged at $100,000 \times g$ for 10 min at 4 °C and the pellet was resuspended in of ice-cold Tris buffer, which was facilitated with the aid of the Polytron homogenizer. The resulting suspension was recentrifuged and the pellet obtained suspended in 50 mM Tris buffer (pH 7.7 at 25 °C) to a volume of 5 mL/g original tissue weight. The whole-brain and striatal membranes were aliquoted into microcentrifuge tubes and stored at −70 °C until needed. Protein concentrations of the rat brain tissues were determined according to the Bradford protein assay using bovine serum albumin as reference standard [18], and protein concentrations for both rat whole-brain and striatal membranes were 6.91 mg/mL and 6.93 mg/mL, respectively.

### 2.5. [³H]-Citalopram Radioligand Binding Assay (SERT)

The method described by Nielsen et al. [4] and Plenge et al. [19], with minor modifications was used. Plant extract (25 μL) was mixed with 50 μL of 0.7 nM [³H]-citalopram and 225 μL of whole-brain membrane suspension. Paroxetine (10 μM) was used for the determination of nonspecific binding. All the samples were incubated for 2 h at 25 °C and filtered under vacuum using glass fiber filters. After 2 h, the radioactivity of the filters containing protein-bound [³H]-citalopram was measured by a liquid scintillation counter (Packard Tri-CARB 2100 TR), using 4 mL filter count as scintillation fluid.

### 2.6. [³H]-DPCPX Binding Assay ($A_1$) and [³H]-NECA Binding Assay ($A_{2A}$)

The degree of binding affinity for plant extracts shown toward adenosine $A_1$ and $A_{2A}$ receptors was determined using radioligand binding assays, as previously described [20–22].

### 2.7. Guanosine Triphosphate (GTP) Shift Assay

Membrane preparation was performed under the same conditions as described above. The GTP shift assay followed a similar method as the $A_1$ AR radioligand binding assay, and 100 μM GTP was added as previously described [23].

## 3. Results

### 3.1. Degree of Affinity (Competitive Binding Assays)

From the three plants, 18 extracts from six solvents were used to test affinity towards SERT in the [³H]-citalopram binding assay, the adenosine $A_1$ receptor in the [³H]DPCPX radioligand binding assay and the adenosine $A_{2A}$ receptor in the [³H]NECA radioligand binding assay (Table 2). Extracts with specific binding percentages of ≤20% at a maximum tested concentration of 1 mg/mL were selected for full screening, in which eight concentrations were analyzed in triplicate (double serial dilution, 0.0078–1 mg/mL).

For the SERT binding assays, only the methanol and acetone extracts of *Hypoxis hemerocallidea* showed relatively poor affinity toward the SERT. For these extracts, [³H]-citalopram showed specific binding values of approximately 60%, an indication that 40% of the plant extract was bound to the SERT. There are no previous reports on plant extracts having affinity for CNS receptors. The extracts of *Leonurus leonurus* and *Gomphocarpus fruticosus* did not indicate activity. Previously, South African plants that have shown binding affinity toward SERT, including *Leonotis leonurus* ethanolic extract [4].

For the adenosine $A_1$ and $A_{2A}$ receptor binding assays, the results indicated that methanol and hexane extracts of *Leonotis leonurus* were the most active, with $rA_1K_i$ values of 0.038 mg/mL and 0.176 mg/mL, respectively. On the other hand, DCM extract of *Gomphocarpus fruticosus* had an $rA_1K_i$ value of 6.46 mg/mL. Coffee is known to be effective for treating a variety of illnesses, particularly lowering the risk of depression [24–27]. Caffeine blocks the adenosine $A_1$ and $A_{2A}$ receptors, and this in turn increases levels of other neurotransmitters in the brain, including dopamine. During the experiment, caffeine showed moderate binding affinity toward both adenosine receptors ($rA_1$ $K_i$ 10.25 mg/mL and $rA_{2A}$ $K_i$ 5.398 mg/mL). Notably, plant extracts from the present study generally had better affinity toward adenosine $A_1$ receptors than caffeine, but not better than the potent and selective adenosine $A_1$ antagonist DPCPX.

**Table 2.** *Leonurus leonotis* (L.) R.Br, [L.l], *Gomphocarpus fruticosus* (L.) W.T. Aiton [G.f] and *Hypoxis hemerocallidea* Fisch. & C.A. Mey [H.h] extracts screened for affinity to the serotonin transporter and adenosine $A_1$/$A_{2A}$ receptors.

| Plant | Solvent | $K_i$ Value ± SD (mg/mL) [a] (Specific Binding (%)) [b] | | | | | | | | |
|---|---|---|---|---|---|---|---|---|---|---|
| | | rA₁ vs. 0.1 nM [³H]DPCPX [c] | | | rA₂A vs. 4 nM [³H]NECA [d] | | | rSERT vs. 0.7 nM [³H]-Citalopram [e] | | |
| | | 0.01 mg/mL | 0.1 mg/mL | 1 mg/mL | 0.01 mg/mL | 0.1 mg/mL | 1 mg/mL | 0.01 mg/mL | 0.1 mg/mL | 1 mg/mL |
| L.l | Water | (91) | (83) | (57) | (97) | (73) | (55) | (94) | (100) | (100) |
| | Methanol | (92) | (29) | (5) | (100) | (93) | (49) | (80) | (100) | (100) |
| | | 0.038 ± 0.009 | | | | | | | | |
| | Acetone | (98) | (20) | (25) | (100) | (68) | (0) | (100) | (100) | (100) |
| | | | | | 0.296 ± 0.069 | | | | | |
| | Dichloromethane | (67) | (21) | (20) | (100) | (89) | (10) | (100) | (100) | (100) |
| | | 0.18 ± 0.009 | | | 0.963 ± 0.016 | | | | | |
| | Petroleum ether | (75) | (31) | (32) | (94) | (100) | (25) | (100) | (100) | (100) |
| | Hexane | (89) | (54) | (19) | (81) | (100) | (76) | (65) | (100) | (100) |
| | | 0.179 ± 0.009 | | | | | | | | |

**Table 2.** *Cont.*

| Plant | Solvent | $K_i$ Value ± SD (mg/mL) [a] (Specific Binding (%)) [b] | | | | | | | | |
|---|---|---|---|---|---|---|---|---|---|---|
| | | rA$_1$ vs. 0.1 nM [3H]DPCPX [c] | | | rA$_{2A}$ vs. 4 nM [3H]NECA [d] | | | rSERT vs. 0.7 nM [3H]-Citalopram [e] | | |
| G.f | Water | (86) | (79) | (56) | (95) | (88) | (94) | (100) | (100) | (100) |
| | Methanol | (93) | (83) | (39) | (100) | (98) | (87) | (100) | (100) | (100) |
| | Acetone | (66) | (64) | (24) | (91) | (84) | (31) | (99) | (100) | (93) |
| | Dichloromethane | (91) (73) (20) 6.83 ± 1.94 | | | (100) | (100) | (69) | (84) | (100) | (100) |
| | Petroleum ether | (92) | (80) | (34) | (100) | (100) | (100) | (100) | (73) | (100) |
| | Hexane | (96) | (93) | (68) | (100) | (100) | (100) | (100) | (100) | (100) |
| H.h | Water | (100) | (100) | (69) | (100) | (100) | (100) | (100) | (100) | (96) |
| | Methanol | (100) | (100) | (62) | (100) | (100) | (99) | (100) | (100) | (69) |
| | Acetone | (100) | (88) | (32) | (100) | (100) | (71) | (100) | (100) | (62) |
| | Dichloromethane | (100) | (100) | (72) | (100) | (100) | (100) | (100) | (100) | (100) |
| | Petroleum ether | (100) | (88) | (49) | (100) | (100) | (100) | (100) | (100) | (100) |
| | Hexane | (100) | (100) | (100) | (100) | (100) | (100) | (100) | (100) | (100) |
| Reference standards | | | | | | | | | | |
| Caffeine | | 10.25 ± 1.437 | | | 5.398 ± 0.019 | | | - | | |
| CPA | | 0.00228 ± 0.0000335 | | | - | | | - | | |
| DPCPX | | 0.000152 ± 0.0000304 | | | - | | | - | | |
| IST | | - | | | 0.00115 ± 0.000346 | | | - | | |
| CB | | - | | | - | | | 0.000486 ± 0.000243 | | |

[a] Inhibition constant ($K_i$, μM) is presented as the mean ± standard deviation (SD), number of repetitions = 3. [b] Specific binding (%) at maximum tested concentration of 100 μM is presented as the mean, number of replicates = 2. [c] Dissociation constant ($K_d$): 0.36 nM [20]. [d] $K_d$: 15.3 nM [21]. [e] $K_d$: 0.84 nM [28] r: rat; SERT: serotonin transporter; [3H]DPCPX: 8-cyclopentyl-1,3-dipropylxanthine, tritiated at dipropyl-2,3-positions, selective adenosine A$_1$ receptor antagonist; [3H]NECA: 5'-N-ethylcarboxamidoadenosine, tritiated at adenine-2,8 position, nonselective adenosine A$_1$, A$_{2A}$, and A$_3$ receptor agonist; [3H]Citalopram: tritiated at the N-methyl group, selective serotonin reuptake inhibitor (SSRI) antidepressant drug, and has been shown to block the serotonin transporter (SERT, 5-HTT); CPD: N$^6$-cyclopentyladenosine; DPCPX: 8-cyclopentyl-1,3-dipropylxanthine (selective adenosine A$_1$ receptor antagonist); IST: istradefylline (selective adenosine A$_2$A receptor antagonist); CB: citalopram bromide (SSRI).

### 3.2. Type of Activity (GTP Shift Assay)

The selected plant extracts were evaluated to establish the type of activity (agonist, inverse agonist, or antagonist) at adenosine A$_1$ receptors (Table 3). It is known that GTP functions by uncoupling the A$_1$ receptors from its G protein, which changes the affinity of A$_1$ receptors from high to low for agonists [29]. By contrasting the binding curves of a plant extract in the presence and absence of GTP, it is possible to determine whether the extract will act as an agonist, an inverse agonist, or an antagonist [30]. An antagonist's binding curve is unaffected by GTP, and the GTP shift value will be close to 1 [31,32]. No significant rightward shift of the binding curve in the presence of GTP was observed, and plant extracts showed GTP shift values of approximately 1 (Supplementary Figure S1). The current results suggested that all the plant extracts act as A$_1$ AR antagonists, and thus, adenosine cannot bind to the said receptor and exert its effects. At presynaptic nerve terminals, A$_1$ ARs play a role in the release of neurotransmitters [33]. Adenosine acts by inhibiting cholinergic transmission, among others via A1 ARs [34]. The cholinergic system has been associated with a number of cognitive functions, for example, learning and memory as well as emotion [35]. Selective adenosine receptor antagonists are being assessed for their antidepressant effects in animal studies and caffeine has been demonstrated to modulate behavior in classical animal models of depression as it is a nonselective adenosine antagonist for A$_1$/A$_{2A}$ receptors [36].

**Table 3.** $A_1$ adenosine radioligand affinities (in the absence and presence of GTP) and the calculated GTP shifts of selected plant extracts.

| Sample | $K_i \pm$ **SEM (mg/mL)** | | |
| --- | --- | --- | --- |
| | **$A_1$ vs. [$^3$H]DPCPX** | **$A_1$ + GPT vs. [$^3$H]DPCPX** | **GTP Shift** |
| *Leonotis leonurus* MeOH | $0.038 \pm 0.006$ | $0.037 \pm 0.007$ | 1 |
| *Leonotis leonurus* DCM | $0.033 \pm 0.010$ | $0.024 \pm 0.011$ | 0.7 |
| *Leonotis leonurus* hexane | $0.179 \pm 0.006$ | $0.209 \pm 0.033$ | 1 |
| *Gomphocarpus fruticosus* DCM | $6.829 \pm 1.120$ | $1.381 \pm 0.247$ | 0.2 |
| CPA ($A_1$ agonist, μM) | $0.00228 \pm 0.0000335$ | $0.035 \pm 0.005$ | 17.5 |
| DPCPX ($A_1$ antagonist, μM) | $0.000152 \pm 0.0000304$ | $0.000122 \pm 0.0000608$ | 0.8 |

All $K_i$ values determined in triplicate and expressed as mean $\pm$ SEM. Rat receptors were used ($A_1$: rat whole-brain membranes). Guanosine triphosphate (GTP) shift assay, where 100 μM GTP was added to the $A_1$ AR radioligand binding assay. GTP shifts calculated by dividing the $K_i$ in the presence of GTP by the $K_i$ in the absence of GTP. Methanol (MeOH) and dichloromethane (DCM).

## 4. Conclusions

The methanol extract of *Leonotis leonurus* was the most active in terms of adenosine $A_1$ receptor binding affinity. Furthermore, this plant extract acted as an antagonist at the adenosine $A_1$ receptor. Notably, its affinity surpassed that of the prototypical adenosine receptor antagonist caffeine. This study is the initial step in evaluating the use of these plants as traditional medicines for managing depression- and anxiety- like ailments. The plant extracts showed promising effects. Given that depression includes various mechanisms, it is important to take note that general conclusions cannot be made for the antidepressant effect of the investigated plants. Exploring other in vitro assays, computational analysis, in vivo studies, and profiling of isolated compounds can allow an in-depth evaluation of the investigated plants.

**Supplementary Materials:** The following supporting information can be downloaded at https://www.mdpi.com/article/10.3390/scipharm91030041/s1.

**Author Contributions:** Conceptualization, A.M., A.O.A. and M.L.; methodology, A.M., H.D.J.v.R. and M.L.; formal analysis, A.M. and H.D.J.v.R.; investigation, A.M. and H.D.J.v.R.; resources, H.D.J.v.R., A.O.A. and M.L.; writing—original draft preparation, A.M.; writing—review and editing, H.D.J.v.R., A.O.A. and M.L.; supervision, A.O.A. and M.L.; project administration, A.O.A. and M.L.; funding acquisition, M.L. All authors have read and agreed to the published version of the manuscript.

**Funding:** This study was supported by the DSI/NWU Preclinical Drug Development Platform (PCDDP), South African Medical Research Council (MRC) self-initiated funding and the National Research Foundation (NRF, UID 129 870).

**Institutional Review Board Statement:** The study protocol was approved by the North-West University Animal Research Ethics Committee (NWU-AnimCareREC) with reference number NWU-00780-22-A5.

**Informed Consent Statement:** Not applicable.

**Data Availability Statement:** All data related to this study are presented in the manuscript.

**Acknowledgments:** We thank Alison Young (horticulturist, University of KwaZulu-Natal Botanical Garden) and McMaster Vambe for assisting with plant collection. We are grateful to Christina Potgieter (Bews Herbarium, NU) and Prin Naidu (A.P. Goossens Herbarium, North-West University) for assisting with the identification of the plants. Sharlene Lowe [(Centre of Excellence for Pharmaceutical Sciences (Pharmacen)), North-West University] is also thanked for her assistance with radioligand binding assays.

**Conflicts of Interest:** We declare no conflict of interest. The National Research Foundation (NRF) had no role in the design of the study, in the collection, analyses, or interpretation of data, in the writing of the manuscript, or in the decision to publish the results. Any opinions, findings and conclusions or recommendations expressed in this publication are those of the authors, and the NRF does not accept any liability in this regard.

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
