# Peer review of "Preliminary Screening of South African Plants for Binding Affinity to the Serotonin Reuptake Transporter and Adenosine A1/A2A Receptors"

_scipharm, doi:10.3390/scipharm91030041_

Round 1

Reviewer 1 Report

The study investigates the potential antidepressant effects of three South African medicinal plants (Gomphocarpus fruticosus, Hypoxis hemerocallidea, and Leonotis leonurus) which have documented traditional uses for mental health conditions but limited scientific evidence. Investigating local plants can provide more accessible and affordable treatment options. The study uses appropriate in vitro methods (radioligand binding assays) to evaluate the selected plant extracts' affinity for two common antidepressant drug targets - the serotonin transporter (SERT) and adenosine A1 and A2A receptors. Six different solvents are used to extract the plants to determine which provides extracts with the highest activity. The more polar solvents methanol and dichloromethane extracted compounds with the most affinity in the assays. Promising activity is demonstrated by the methanol and acetone extracts of Hypoxis hemerocallidea for SERT binding, and the methanol and hexane extracts of Leonotis leonurus for adenosine A1 receptor binding. The extracts act as adenosine A1 receptor antagonists. The study provides initial evidence on potential mechanisms of antidepressant action for these plants. Further evaluation can be guided by these results.

 The following deficiencies are detected in this study:

·      Only evaluating affinity for SERT and adenosine receptors limits interpretation of antidepressant potential, as depression involves multiple mechanisms. Testing other common targets would provide a more complete understanding.

·      While in vitro binding assays indicate affinity, further functional assays showing activation/inhibition of the receptor are needed to confirm activity.

·      The crude extracts contain many compounds, so the specific active components are unknown. Activity-guided fractionation and isolation of compounds would identify the actives.

·      The study lacks any in vivo antidepressant testing in animal models to demonstrate pharmacological activity in a living system.

·      Limited replicates (triplicates) were performed for generating the quantitative data. More replicates would improve statistical power.

·      Computational approaches like molecular docking could help predict potent plant compounds binding to the receptors.

Please review for any spelling or grammatical errors.

Author Response

Please see the attachment. Please take note that details response is located in file name 'Reviewer 1 response

Reviewer 2 Report

In this present manuscript titled: "Antidepressant effect of three South African plants based on their affinity for serotonin transporter and adenosine receptors”, Andisiwe et al. performed extraction with different solvents from Gomphocarpus fruticosus , Hypoxis hemerocallidea, and Leonotis leonurus and evaluated the affinity for serotonin transporter and adenosine receptors. Overall, the manuscript looks good. A few edits, improvements, and experiments are needed before publishing. Authors should address the following question before publication:

·         The phytochemical profiles of the plant extracts are important. It’s good to provide LCMS profiles of the isolated extracts to understand what molecules are present in those fractions. And by checking the previous literature mention the probable composition of molecules which could be responsible for the efficacy.

·         The cAMP Accumulation Assay is an important assay to perform to jump to conclusions regarding the antagonistic properties of the extracts. Evaluate the effects of the extracts on downstream cAMP formation.

·         The following line needs to be modified in the Abstract section. "Six solvents (water, methanol, acetone, dichloromethane, petroleum ether, and hexane) were used to extract the selected plants using in vitro binding assays." One cannot use an in vitro binding assay for extraction.

·         In the abstract section Line 23: The more polar solvents, methanol and dichloromethane, were more active regarding binding affinity. One cannot claim that only solvents have binding affinity.

·         Double check the paragraph "2.5[3. H]-Citalopram binding assay (SERT)". A repetition of the same lines is found. Remove the unnecessary lines (Lines 123–129).

·         Don’t use the term "[3H]-citalopram binding assay," as it is a competitive binding assay with 3H-citalopram (a radioligand). That’s why it's recommended to use the "Radioligand Binding Assay.".

·         Use [3H]-Citalopram instead of [3H]-Citalopram. Make sure to use [3H] for tritiated hydrogen in the manuscript.

·         "Caffeine is an alkaloid, present in two major species (Coffea arabica and C. canephora) of coffee plants [24]". This line is irrelevant to the current topic. Authors could rearrange the lines to make the paragraph more relevant, e.g., "Coffee is known to be effective for treating a variety of illnesses, and studies have found that drinking coffee could be associated with a lower risk of depression. (Aust N Z J Psychiatry. 2016 Mar;50 (3):228–42. doi: 10.1177/0004867415603131.) Caffeine blocks the receptor adenosine, and this increases levels of other neurotransmitters in the brain that regulate your energy levels, including dopamine. During the experiment, Caffeine showed moderate binding affinity toward both adenosine receptors (rA1 Ki 10.25 mg/ml and rA2A Ki 5.398 mg/ml)".

·         Mention the protein concentration you used to perform the binding assays.

·         Change the term unspecific to nonspecific binding.

Round 2

Reviewer 2 Report

This Article reports “Preliminary Screening of South African plants for binding affinity to the serotonin reuptake transporter and adenosine A1/A2A receptors”

I recommend publishing after minor revisions that should be addressed.

I wouldn't recommend using the terminology "profiling of plant extracts" in the abstract section because it means in this manuscript the authors performed in-depth HPLC/LCMS analysis and identified key components, which has not been done in this manuscript.

The authors mentioned that the DCM extract of Gomphocarpus fruticosus has a rA1Ki value of 6.46 µM (Page 4, Line 163). I would suggest changing it to mg/mL to maintain the consistency of the writing. In most cases, to get the binding efficacy of any unknown plant extract, it is recommended to use mg/mL unit. I wonder, without knowing the MW of the DCM extract, how the authors would identify the Ki value in µM.
